# The 30-year evolution of oral cholera vaccines: A case study of a collaborative network alternative innovation model

**Kaitlin Large** *, **Adrian Alonso Ruiz**, Iulia Slovenski, Marcela Vieira, Suerie Moon

Global Health Centre, Geneva Graduate Institute, Genève, Switzerland

* kaitlin.large@graduateinstitute.ch

## Abstract

Cholera outbreaks have been rapidly increasing around the world. While long-term cholera prevention and control measures rely on improvements in water, sanitation, and hygiene, oral cholera vaccines (OCVs) are used for prevention and control in the short-to-medium term. OCVs lack the market incentives available in other more profitable disease areas. The development of OCVs was made possible through an alternative innovation model, which sustained innovation across multiple generations of the product for more than three decades. To examine how this alternative innovation model worked, we conducted 18 semi-structured interviews with key stakeholders related to the development of several OCVs, including "Dukoral", "Shanchol", and "Euvichol-Plus", as well as additional cholera vaccines currently under development. Interview data was analyzed thematically according to the resources used by organizations (including funding, knowledge, relationships, and manufacturing), and the practices they implemented during each stage of the R&D process (including knowledge management and intellectual property strategies, approaches to transparency, and global access strategies). Next, we created participatory network maps to illustrate the structure of the relationships between stakeholders and how these evolved over time. We found that a core group of stakeholders were able to influence policies to promote the use of OCVs, and successfully develop, finance, and obtain WHO Prequalification for safe, effective, and affordable OCVs for global procurement and distribution. The evolution of OCVs demonstrates how a collaborative network innovation model can successfully develop new pharmaceutical products that are affordable and well-suited for use in context. This model could be applied to other areas of pharmaceutical innovation, such as pandemic preparedness, for more equitable outcomes for global public health.

## I. Introduction

In recent years, the incidence of cholera outbreaks has been rapidly increasing around the world [1]. In 2023, the number of cholera cases reported to the World Health Organization (WHO) surpassed the year before, as 4,000 cholera deaths and 667,000 cases were reported globally, and the resurgence of cholera has been classified as a 'grade 3 emergency', the WHO's

**Data availability statement:** The data that support the findings of this study are openly available in Zenodo at http://doi.org/10.5281/zenodo.13373057.

**Funding:** This work was supported by a Swiss National Science Foundation PRIMA Grant (179842) (SM). https://www.snf.ch/en. The funder had no role in study design, data collection and analysis, decision to publish, or preparation of the manuscript.

**Competing interests:** The authors have declared that no competing interests exist.

highest internal health emergency level [2]. While long-term cholera prevention and control measures rely on improvements in water, sanitation, and hygiene (WASH), since 2010, the WHO has also recommended the use of oral cholera vaccines (OCVs) as a complementary cholera prevention and control measure in the short-to-medium term, in both endemic countries and during cholera outbreaks [3]. While the value of using OCVs was previously contested, studies demonstrated that when considering vaccine herd protection, OCV programs were cost-effective [4]. Therefore, the innovation model used to develop OCVs warrants further attention particularly given the unique dynamics of the cholera vaccine market. The OCV story is more than just an example of a single product developed for global health, but rather a story of sustained innovation across multiple generations of the product over more than three decades, making it a particularly interesting case to study.

Cholera is a bacterial disease which can lead to severe dehydration and death if left untreated, and disproportionately affects populations living in poverty [1]. Severe cholera has a fatality rate exceeding 50% if left untreated [5]. The market for cholera vaccines faces challenges typical of neglected disease markets, such as a lack of financial viability [6]. In addition, much like the pandemic preparedness market, demand forecasting is difficult and vaccines are used as acute treatments [7]. For example, at least one confirmed cholera case locally acquired is considered an outbreak [8] leading to constant fluctuations in vaccine demand. Also, the fact that OCVs are priced low and require bulk sales to generate profit make them unattractive to mainstream commercial drug developers [9]. As a result, the market for travel-related immunizations for OCVs is limited in size and only a few countries manufacture OCVs for national use: Vietnam, China, Bangladesh [10], and the United States [11]. Instead, the market is composed of international agencies who procure OCVs and distribute them via the global OCV stockpile, which is entirely donor-funded by Gavi, the Vaccine Alliance, and vaccines must receive WHO Prequalification (PQ) to be eligible for procurement. The WHO's International Coordinating Group (ICG) on Vaccine Provision manages the emergency stockpile of doses designated for outbreaks and other emergency situations, while the remainder of the doses in the stockpile are allocated to countries by Gavi, for use in multi-year vaccination plans that are largely preventive in nature.

In the midst of increasing cholera outbreaks and therefore demand for OCVs, the global OCV stockpile has failed to procure adequate vaccine supply due to the limited funding available to purchase cholera vaccines, and the resulting limited production by suppliers [12]. Specifically, cholera vaccines are complex to produce and have certain lead times for manufacturing, which require clear demand forecasting in order to adjust the supply capacity to the stated need. This has led to severe global OCV shortages, further building upon the persistent problem of a lack of OCV supply, ever since the first vaccine received WHO PQ. For example, in November 2023, only 45% of the 65 million OCV doses requested had been approved and allocated to countries, with the shortage projected to last until 2025 [13,14]. In addition, vaccine campaigns temporarily suspended the recommended two-dose vaccine regimen in an effort to ration doses and have transitioned to a single-dose strategy instead [15].

As of early 2024, the global OCV stockpile was being supplied by only one vaccine manufacturer – EuBiologics, a small biotechnology company from the Republic of Korea (ROK), which produces the OCVs "Euvichol-Plus", and a newly simplified formulation "Euvichol-S", which received WHO PQ on 12 April 2024 [16]. The development of Euvichol-Plus and Euvichol-S built upon each of their OCV predecessors, and together, the evolution of OCVs tells a story of a small network of actors who not only made a compelling use-case for cholera vaccination, but also worked to generate political will and mobilize the necessary resources for vaccine development along the way.

While the innovation model used by product development partnerships (PDPs) has been examined in academic literature, there is not, to our knowledge, significant scholarship on a collaborative network model of pharmaceutical innovation that is perhaps less viable since it is not led by one organization, but rather a network of multiple organizations. While initiatives like the COVID Moonshot project appeared to be novel in their use of a consortium approach to drug development, the evolution of OCV development and its collaborative network innovation model suggests that this approach is not entirely new. Furthermore, while there is literature on drug development (or previously the lack thereof) within the neglected diseases [6,17,18] and pandemic preparedness niches [19,20], there is a lack of literature on products or diseases that may reflect the characteristics of both niches, such as cholera does. The aim of this in-depth case study is to examine how an alternative innovation model enabled OCVs to progress from inception to procurement and to analyze how a network of key stakeholders developed vaccines for global public health.

## II.  Methods

This case study is part of a larger five-year research project on alternative models of pharmaceutical innovation, which aimed to deepen understanding of the political, economic, scientific, and organizational factors required to implement alternative innovation models that deliver both innovation and globally affordable medicines. We defined the mainstream innovation model as one in which "a commercial profit-maximizing firm conducts the later stages of research and development (R&D) (e.g., preclinical to clinical trials) and brings a product to market. Competition between companies and market incentives influence which diseases or technologies the firm prioritizes; how it manages knowledge such as data and intellectual property; and its strategies for obtaining regulatory approval, production, marketing, distribution, and pricing. Usually, this firm is based in a high-income country" [21]. Alternative innovation models differ from the mainstream along one or more of these areas.

Within the project, we applied the concept of complex adaptive systems (CAS) to the pharmaceutical innovation system to facilitate our analysis of it, which yielded a Complex Adaptive Pharmaceutical Innovation System (CAPIS). A detailed methodology and conceptual framework for the broader research project is available from the authors upon request. Within the CAPIS, we identified four niches for pharmaceutical R&D that have experienced clear instances of market and/or system failure, and where we have found efforts to adopt alternative innovation models: rare diseases, neglected diseases, pandemic preparedness/ biosecurity, and antibiotics. Most closely related to the neglected diseases and pandemic preparedness niches, the data corpus for the present case study on the evolution of OCVs consists of semi-structured interviews, gray literature in the form of organizational reports and news articles, and academic literature.

### A.  Participant recruitment

We contacted four interview participants through purposive sampling and recruited an additional 16 participants through snowball sampling. Of the 20 potential participants contacted via email, 18 participants agreed to participate in semi-structured interviews for the case study. Interviewees represented funders (those who provide push or pull funding for R&D, n = 7), implementers (those who conduct R&D activities, n = 7), and facilitators (those who help to facilitate the R&D process in some way, n = 4) associated with the development of the OCVs "Dukoral", "Shanchol", and "Euvichol-Plus", as well as those associated with additional cholera vaccines currently under development, which are tangentially related to the same network of stakeholders.

## B. Data collection

Semi-structured interviews were conducted by one researcher with 18 participants from 11 different organizations between 11 April 2023 and 27 September 2023. Interviews lasted between 60–90 minutes, and eight took place online via video conference platforms while nine were conducted in-person during a site visit to the International Vaccine Institute (IVI)'s, EuBiologics', and the RIGHT Foundation's headquarters in Seoul, Republic of Korea (ROK), from 19 June 2023 to 23 June 2023. Finally, one interview was conducted via email. Interview guides were tailored to each type of interviewee (whether funder, implementer, or facilitator), and questions asked pertained to the practices of the organization itself, its role in the development of the cholera vaccine in question, and broader questions about the overall pharmaceutical innovation system in which the organization was embedded. Interviews were conducted in English. With the informed consent of each interviewee, interviews were recorded and transcribed with Otter.ai software, and transcripts and quotes were edited for clarity. Due to the sensitivity of the information being shared, all quotes were anonymized to protect the participants' privacy. Each interviewee is referred to by their identifier (i.e., I_1, 2, 3, …). None of the interviewees reported any conflicts of interest related to their participation in the research.

## C. Ethics statement

Ethics approval for the study was obtained from the Geneva Graduate Institute's Ethics Review Committee on 16 March 2020. All methods were performed in accordance with the relevant guidelines and regulations of the Declaration of Helsinki. Informed formal written consent for participation was obtained from all study participants. Written informed consent for publication of individual details and quotes used within the manuscript was also obtained from all relevant study participants.

## D. Data analysis

Interview data was analyzed thematically [22,23] according to the CAPIS analytical framework developed by the research team for the broader project. One researcher analyzed all interview transcripts and identified the resources used by various stakeholders (including funding, knowledge, relationships, and manufacturing), and the organizational practices implemented (including knowledge management and intellectual property strategies, approaches to transparency, and global access strategies) during each stage of the R&D process. The research team iteratively discussed the OCV case findings, comparing the OCV case to other initiatives examined by the project.

A recurrent theme that emerged from the qualitative interview data was the importance of relationships between a relatively small group of actors throughout the more than 30-year evolution of OCV development. In tracing this story over multiple decades, it became clear that mapping the evolving relationships between actors would help facilitate further understanding of the innovation model. Therefore, we created participatory network maps to illustrate the structure of the relationships between the stakeholders involved in the development of the OCVs examined, and how their relationships evolved over time and in meaning. We believe this analytical approach provides the reader with a deeper understanding of the structure of the OCV development network and the interpersonal relationships embedded within it, which is key to explaining how this long-term alternative innovation model worked. All interviewees had an opportunity to review a draft of the manuscript and provide comments on it.

## III. Results

The evolution of OCVs is an innovation story which encompasses the continuous expansion of a network of individuals, largely based at not-for-profit entities, who developed not only the first OCV, but also subsequent improvements to the product to make it more affordable, easier to transport, and easier to administer, over a more than 30-year timeline. To illustrate this evolution, the results section is structured from the oldest OCV, Dukoral, to the most recent OCV at the conclusion of the research project, Euvichol-Plus, and briefly discusses the other cholera vaccines currently under development. The network of actors involved includes a combination of stakeholders from the Global North and Global South, and a wide range of organization types (academia, intergovernmental organizations, state-owned pharmaceutical companies, foundations, and private firms).

In addition, the research project has identified four types of resources that all organizations need to develop new medicines: funding, knowledge, ability to manufacture, and relationships. With respect to each vaccine discussed in the following sections, our research findings also seek to examine how these resources were mobilized by the network of actors and contributed to their ability to develop each vaccine. We have also used a participatory network map in each section to illustrate the relationships that were formed between actors during the development of each vaccine, and more broadly, how these relationships and the larger cholera vaccine ecosystem evolved over time. Finally, in the last section, we briefly discuss the implications of the cholera vaccine market being entirely donor-funded, and how this relates to the shortages being experienced today. However, a full explanation of the cholera vaccine shortages is beyond the scope of this paper.

Fig 1 provides a timeline overview of the cholera vaccines discussed in the results section. This paper does not discuss all cholera vaccines under development, but rather those that are tangentially linked to the original OCV, Dukoral. The timeline includes the name of the vaccine, the year it was licensed and/or received WHO PQ, and the developer's country. Next, Table 1 expands upon the timeline and provides additional information about the developers of each vaccine, the external funders who funded the vaccine's development, the product characteristics of the vaccine, and the current availability or stage of development of the vaccine.

### A. Dukoral

The first OCV ever created was Dukoral, developed by Professor Jan Holmgren's team at the University of Gothenburg in collaboration with the Swedish National Bacteriological Laboratory (SBL). Before the invention of OCVs, injectable cholera vaccines had long been used in vaccination campaigns. However, the WHO stopped recommending the use of injectable cholera vaccines as early as the 1970s, due to their limited prolonged efficacy and risk of

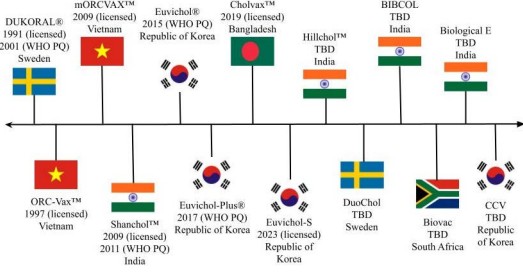

**Fig 1. Cholera vaccine development timeline.**

**Table 1. Overview of cholera vaccines.**

| Vaccine name | Developers | External funders | Product characteristics | Availability/stage of development |
|---|---|---|---|---|
| DUKORAL | University of Gothenburg (Sweden), SBL Vaccines (Sweden), Valneva Sweden AB (Sweden) | Swedish Research Council (SRC), SIDA, USAID | Contains inactivated *Vibrio cholerae* O1 bacteria and the recombinant non-toxic B-subunit of the cholera toxin (rCTB)2. Administered with 150 ml of water and a buffer solution [24]. | Available and WHO prequalified. |
| ORC-Vax | VaBiotech (Vietnam) | SIDA | Contains bivalent vaccine with *V. cholerae* O139 [25]. | No longer in production. |
| mORCVAX | VaBiotech (Vietnam), IVI (ROK) | BMGF | Reformulated the ORC-Vax vaccine and expressed the antigen content in ELISA units [25]. | Licensed and available for use in Vietnam. |
| Shanchol | IVI (ROK), Shantha Biotechnics Ltd. (known as Sanofi Healthcare India Private Ltd (SHIPL) since 2019 and part of the Sanofi Group since 2011) (India) | BMGF | Contains killed whole-cell O1 and O139 serogroups without B subunit cholera toxin [25]. | Previously available and WHO prequalified. No longer in production as of 2023. |
| Euvichol/ Euvichol-Plus | IVI (ROK), EuBiologics (ROK) | BMGF, Korea-Seoul Life Science Fund, Green Cross Corporation, Shinhan-K2 Investment Partners, Global Health Investment Fund, Korea Investment Global Frontier Fund | Same formulation as Shanchol, immune non-inferiority study conducted in 2013, results published in 2015 [25,26]. Euvichol-Plus is produced in a plastic tube instead of a glass vial. | Available and WHO prequalified. |
| Cholvax | IVI (ROK), Incepta Vaccine Ltd (Bangladesh) | Information unavailable | Same formulation as Shanchol. | Available in Bangladesh, not yet WHO prequalified. |
| Euvichol-S | IVI (ROK), EuBiologics (ROK) | BMGF | A simplified formulation of Euvichol-Plus which reduces components from five to two and the inactivation process from two to one [27]. | Available and WHO prequalified. |
| Hillchol | University of Gothenburg (Sweden), Hilleman Labs (Singapore), Bharat Biotech (India) | SRC, Vinnova (Sweden), BMGF | Single component, inactivated V. cholerae O1 Hikojima El Tor strain facilitating low cost and large quantity manufacturing (I_2) | Licensed and available for use in India. |
| DuoChol | University of Gothenburg (Sweden), IVI (ROK) | Wellcome Trust, SIDA | Contains a dry formulation inactivated bacterial whole cell/cholera toxin B subunit with a similar composition as DUKORAL [28]. | Currently in phase I clinical development. |
| Technology Transfer (OCV) | IVI (ROK), Bharat Immunologicals and Biologicals Corporation Limited (BIBCOL) (India) | Information unavailable | Same formulation as Shanchol | Technology transfer to be completed by 2025. |
| Technology Transfer (OCV) | IVI (ROK), Biovac Institute (South Africa) | Wellcome Trust, BMGF | Same formulation as Euvichol-Simplified | Technology transfer process has commenced, clinical trials expected in 2024. |
| Technology Transfer (OCV) | IVI (ROK), Biological E (India) | BMGF | Same formulation as Euvichol-Simplified | Technology transfer to be completed by 2025. |
| Cholera conjugate vaccine (CCV) | IVI (ROK), Harvard/Massachusetts General Hospital (USA), EuBiologics (ROK) | Wellcome Trust, RIGHT Foundation, Open Philanthropy, US National Institutes of Health | Cholera conjugate vaccine | Toxicological analysis of CCV in animals completed; preparing for phase 1 evaluation of CCV in humans. |

adverse side effects [29] (I_6). At the time, the mainstream public health response instead shifted to focusing on the use of oral rehydration solution (ORS) to treat cholera patients in the short-term and emphasized the need to improve WASH for long-term cholera prevention and control (I_6).

While the public health response shifted away from vaccination and toward alternative treatment and control measures, simultaneously scientists were beginning to develop an improved understanding of mucosal immune responses in the late 1970s (I_6). Holmgren and

his team at the University of Gothenburg, including Professor Ann-Mari Svennerholm as his primary collaborator, studied the protective immune mechanisms in cholera and discovered a more effective way of cholera vaccination – OCVs. Specifically, they found that "oral immunization with a combined inactivated whole-cell/B-subunit (WC-BS) vaccine should be an effective way to induce protective immunity against cholera" [29]. The results obtained from their studies became the data composition used for Dukoral, and they began translational work together with SBL in Sweden.

With the University of Gothenburg's collaboration with SBL underway, Holmgren also began contacting his long-standing scientific personal contacts, including those who were scientists working at the International Centre for Diarrheal Disease Research, Bangladesh (icddr,b), who focused on cholera research. One of these scientists was Dr. William Greenough, who had since become the Director-General of icddr,b, and through whom Holmgren established a formal collaboration with the organization in 1978 (I_2). Working with Greenough were two scientists with cholera expertise – Dr. David Sack and Dr. Roger Glass, whom Holmgren also began collaborating with on the OCV development.

Holmgren was one of the first board members of the icddr,b and was also a member of the Swedish International Development Cooperation Agency (SIDA) advisory board. Through his connections, he was able to convince the Government of Sweden to become a donor to the icddr,b (I_2). Once the OCV was ready for phase 3 clinical trials in 1985, the icddr,b conducted a phase 3 trial of two OCV formulations in Bangladesh, which was led by Dr. John Clemens and Dr. David Sack of the icddr,b [29]. The results from the trial demonstrated that both formulations were protective against cholera (efficacy was 85% for the combined inactivated whole-cell/B-subunit formulation (WC-BS) and 58% for the whole-cell only (WC) formulation over the first 4–6 months, and 50%–55% for both vaccines over 3 years of follow-up) [29].

After a severe outbreak of cholera in Peru, a second phase 3 trial was conducted with military recruits in Peru in 1992 [30], which confirmed the results from Bangladesh [30]. Using the more sophisticated WC-BS formulation, Dukoral was successfully licensed in Sweden and other Scandinavian countries in 1991, and in 1993 internationally [29,31] (I_2).

As a result of Holmgren's professional connections, Dukoral was able to be produced with limited funding consisting of entirely public and philanthropic sources:

*"Dukoral [was developed] with extremely little money…Basically, it was done with the normal university grants together with the support from SBL, which was not insignificant…then when it came to needing big money for undertaking the large phase three trial in Bangladesh, many donors of* icddr,b *contributed, but in particular, USAID, which was the major funder of this extremely large and ambitious field trial" (I_2)*

Fig 2 provides the participatory network map legend. Fig 3 presents the participatory network map for Dukoral, which illustrates how these individuals and entities originally became connected to one another, and thus created the fabric for the cholera vaccine ecosystem that continued to evolve over the next several decades. The length and ombre color of certain lines, as well as the size and placement of shapes do not hold additional meaning when interpreting the maps, but instead are automated features of the software program used to design them. However, for each vaccine in question, the maps capture direct transfers of funding between donors and recipient, as well as knowledge sharing and manufacturing flows.

Dukoral paved the way for the next generation of OCVs and acted as the foundation for those that followed. Holmgren and his team chose not to patent the original vaccine technology since it was a vaccine for global health, and instead published their scientific results openly. In addition, they used a dual-market strategy to earn revenue on the traveler's market

| Shape Key | | Color Key | |
|---|---|---|---|
| Shape | Type | Color | Resource |
| ▼ | Individual | 🟥 | Means of manufacturing |
| ● | Entity | 🟩 | Funding |
| | | 🟦 | Knowledge |

**Fig 2. Participatory network map legend.**

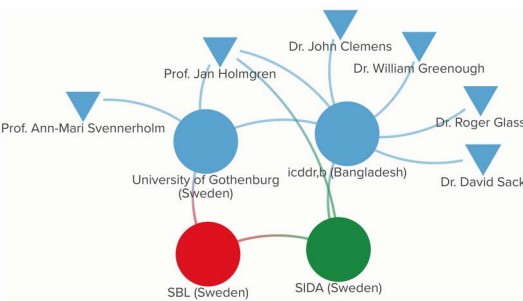

**Fig 3. Participatory network map for Dukoral.**

which also supported Dukoral's ability to be approved and made available on the public market:

> *"[The OCV] is a vaccine for the poorest people in the poorest countries. Therefore, it has had quite little commercial interest to companies. Dukoral also through the B subunit held as an indication protection against diarrhea caused by enterotoxigenic E. coli, ETEC diarrhea, or as it's often called, traveler's diarrhea. Therefore, Dukoral found a market and has been largely used as a traveler's vaccine. Had it not been for that indication, it's unlikely that any OCV would have come to the market, at least at that time. So, the ETEC indication probably was very important for getting OCVs to the market" (I_2)*

In 1997, Dr. Dominique Legros, who was the Head of Country Office at the MSF affiliated Epicentre Research Institute in Uganda, Dr. Bernard Ivanoff, who was a Medical Officer with the WHO, and their colleagues, conducted the first feasibility study of using Dukoral in a mass vaccination campaign in a refugee setting [32]. This study demonstrated that mass vaccination of a refugee population with Dukoral was feasible, but also emphasized the need to create a global OCV stockpile before use of OCV could be recommended [32]. Dr. Legros later went on to head WHO's cholera program, and in collaboration with Dr. Ivanoff among others, was instrumental in further developing the policy environment to be receptive to OCV recommendation and usage.

## B. ORC-Vax/mORC-Vax

The second OCV called ORC-Vax was developed by VaBiotech (a Vietnamese state-owned biotechnology company) for national use in Vietnam. Like each of the OCVs that follow, the

development of ORC-Vax directly built upon Dukoral. The Government of Sweden had been a long-standing donor country to Vietnam, and the SBL scientists had been collaborating directly with Professor Dang Duc Trach of the National Institute of Hygiene and Epidemiology (NIHE) in Vietnam (I_2). Encouraged by the results of the phase 3 trials of Dukoral in Bangladesh, Trach was interested in producing OCVs locally (I_2). After contacting Holmgren, the SBL agreed to provide the technology transfer to Trach's team in Vietnam free of charge.

In parallel, Trach asked John Clemens, the epidemiologist with icddr,b who had led Dukoral's phase 3 trial in Bangladesh, to help him reformulate the vaccine for use in Vietnam (I_2). Vietnam did not have the technology necessary to produce the WC-BS formulation of the OCV which was used in Dukoral and wanted to produce a lower cost alternative. Therefore, clinical trials were conducted with the WC-only formulation of the OCV, and this formulation demonstrated 66% protective efficacy (I_2) [29]. A few years later, when a new cholera strain (*Vibrio cholerae* O139) [33] was detected in Bangladesh and India, Holmgren's team worked with Trach's team once again to help them reformulate the vaccine to include the new strain. This new formulation was licensed as ORC-Vax in 1997 for use in Vietnam, and production was transferred from the NIHE to the newly created VaBiotech facilities [29,31].

Fig 4 presents the participatory network map for ORC-Vax, which highlights the actors who were involved in its development and uses grayscale shading to signify those within the existing network who were not directly involved.

The year 1997 also marked the establishment of the International Vaccine Institute (IVI), whose creation was proposed by the United Nations Development Programme (UNDP) for the purpose of discovering, developing, and delivering safe, effective, and affordable vaccines for global public health [34]. After leaving the icddr,b and spending a few years at the US National Institutes of Health (NIH), Clemens was appointed as IVI's first Director-General in 1999 [34]. During this time, the official WHO position remained that cholera vaccination was not part of a public health response (and instead the focus remained on ORS and long-term WASH efforts), despite the high protective efficacy rates of OCVs such as Dukoral (which was WHO Prequalified) and ORC-Vax (which was licensed only for use in Vietnam) (I_4). In part, this was due to the requirement for a large amount of buffer (150ml) prior to administration of Dukoral, which was not considered pragmatic for use in outbreak settings. Interviewee I_4 further described the tension between public health professionals:

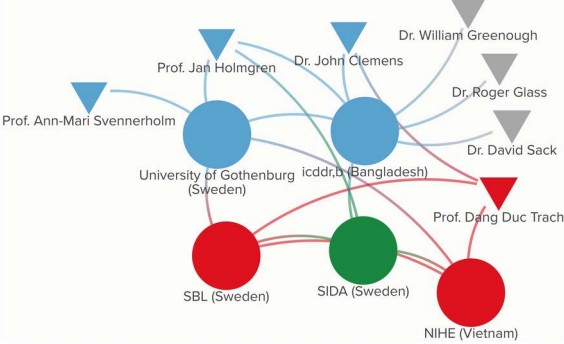

**Fig 4. Participatory network map for ORC-Vax.**

*"At the turn of the century, WHO was really focused more on WASH interventions and treatment options, and not on the use of OCV. Within WHO, the Initiative for Vaccine Research (IVR) emphasized the conduct of research in support of OCVs. Using Dukoral, IVI, with funding from IVR, conducted a first study in an African population in Mozambique with high rates of HIV-infection and endemic cholera which showed the public health benefit of the vaccine in this setting. A second study, funded by the Bill and Melinda Gates Foundation (BMGF) to IVR, evaluated the benefit of preventive campaigns in Zanzibar where annually, cholera 'hotspots' occurred across the islands" (I_4)[35,36]*

The grant for the effectiveness study with IVI was part of the BMGF's "Diseases of the Most Impoverished" (DOMI) program. The BMGF had been newly founded in the year 2000, and shortly thereafter awarded IVI a $40 million grant to generate scientific evidence on the burden of cholera, typhoid fever, and shigellosis, and to develop new and improved vaccines against the diseases [34,37]. Interviewee I_4 explained that part of the program, which ran for approximately seven years, included conducting phase 4 vaccine effectiveness studies of Dukoral in Mozambique and Zanzibar, and demonstrated 78% protective efficacy against cholera outbreaks 4–5 months after vaccination in the Mozambique study [36] and 79% protective efficacy against cholera outbreaks 1 year and 2 months after vaccination in the Zanzibar study [38].

Using this evidence, cholera experts were able to make a strong argument for the use of OCVs and convince the WHO to offer Prequalification (PQ) so that OCVs could be purchased and distributed by international procurement agencies. As a result, in 2001 Dukoral earned WHO PQ. However, it was rarely procured due to its relatively high price of between $3–$6 per dose (as a result of its complex formulation), and bicarbonate buffer co-administration requirements, making it unsuitable for resource-constrained settings [39]. In addition to the cost and administration requirements of Dukoral, at the time certain segments of the WHO feared that introducing OCVs would detract from their continued investment in other treatment methods and in WASH (I_2).

In 2006, the BMGF awarded IVI a grant to launch the Cholera Vaccine Initiative (CHOVI) to develop and introduce new, affordable OCVs in countries affected by cholera [34]. The product profile of the ORC-Vax vaccine proved to be more suitable for resource constrained settings due to its simpler technology, affordability, and lack of requirement to co-administer it with a buffer. Thus, IVI worked with VaBiotech to reformulate the vaccine to be compatible with modern Good Manufacturing Practice (GMP) standards and therefore, potentially eligible for procurement by international procurement agencies (I_2). After successful clinical trials, in 2009, the reformulated vaccine was licensed in Vietnam as mORC-Vax [29] (I_2). Fig 5 presents the participatory network map for mORC-Vax.

As one interviewee described:

*"[OCV development] was made possible frankly, by partnerships and friendships…The strength of having John Clemens at IVI, Bernard Ivanoff at WHO, Roger Glass at CDC, Trach at VaBiotech, and Jan Holmgren, the original cholera vaccine developer, really energized and engineered that whole development program. It was funded by the BMGF, but without them, I don't think it would have happened" (I_4)*

While mORC-Vax had been successfully used throughout Vietnam, the vaccine was not eligible to receive WHO PQ because the Vietnamese national regulatory authority (NRA) was not a WHO-listed Authority operating at maturity level 3 or above [29,40]. Therefore, the BMGF grant required IVI to develop and transfer the OCV technology to two vaccine manufacturers

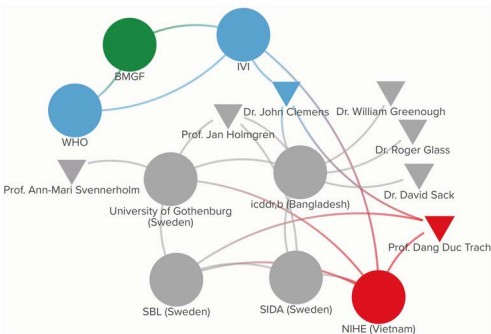

**Fig 5. Participatory network map for mORC-Vax.**

whose NRA already met these requirements, in order to ensure that the OCV could be eligible for WHO PQ and procurement by international agencies (I_4). Beginning with mORC-Vax, IVI began playing an instrumental role as an intermediary entity to help successfully facilitate technology transfer and ensure that the OCVs developed met the characteristics necessary. As Interviewee I_1 explained:

> *"That's often the kind of space IVI fills, certainly for cholera vaccines – kind of that intermediary. IVI didn't invent the vaccine, but they enabled it to become a product that would meet international standards" (I_1)*

## C. Shanchol

As India's NRA met the requirements necessary for WHO PQ, IVI engaged in their first technology transfer per their grant agreement with the BMGF, with Shantha Biotechnics in 2006. Together, IVI and Shantha signed a "Global Access Strategy Agreement" for the production, licensing, and distribution of the OCV mORC-Vax, which was developed by IVI in partnership with VaBiotech (I_16).

Interviewee I_2 explains how the process began, stating:

> *"Dr. Clemens took the initiative to transfer the same refined technology used by [mORC-Vax] to India. He met with a young vaccine company Shantha Biotechnic. Vietnam produced the bulk of the vaccine, and it was initially only fill-finished in India. But India had a regulatory agency which was approved by the WHO" (I_2)*

The technology transfer was led by Clemens with guidance from Ivanoff of WHO [29,31]. Interviewee I_2 explains how Ivanoff played a pivotal role in facilitating the technology transfer:

> *"Dr. Bernard Ivanoff was an important person at the WHO, and was, in a way, living through the problems to get the WHO to accept OCVs...He was also very instrumental in convincing Vietnam to transfer technology to Shantha. I think his diplomatic skills were pivotal to leading to that tech transfer from Vietnam to India" (I_2)*

The technology transfer process was financially supported by the BMGF. As IVI had assisted VaBiotech with the reformulation process for mORC-Vax, IVI possessed "know-how" for the specific strains of inactivated bacteria used in the OCV formulation, which they licensed to Shantha (I_1). At the time, the BMGF did not yet have their own global access terms in place

for grantees, so instead, IVI created and implemented their own global access agreement with Shantha. As interviewee I_4 explained:

> "If the BMGF were doing that grant today, it would be very involved in the [global access] agreement and would actually lay out conditions that the BMGF's market dynamics group would be involved in, so it becomes more complicated. But in those days, it was a requirement of IVI to get global access. They had the IP, which generally the BMGF doesn't want or need. Then, in the tech transfers, IVI made sure that they had global access, which meant volumes and pricing" (I_4)

After successfully completing the technology transfer, IVI then partnered with the National Institute of Cholera and Enteric Diseases (NICED) in India to conduct clinical trials and demonstrated that two doses of the vaccine provided protective efficacy of 65% over a five-year period [41]. As a result, Shanchol was licensed in India in 2009 and received WHO PQ in 2011. Shanchol became available for procurement and distribution by international agencies at a price of $1.85 per dose, a substantial improvement from the $3–$6 per dose price of Dukoral (I_16) [31]. This was made possible by IVI negotiating global access and pricing before the vaccine was licensed. Fig 6 presents the participatory network map for Shanchol.

**Changing cholera ecosystem.** Shortly after Shanchol was licensed, there was a catastrophic earthquake in Haiti in 2010 that killed over 200,000 people and displaced more than 1 million. Soon after the earthquake, there was a huge cholera outbreak with over 820,000 confirmed cases and nearly 10,000 deaths [42]. This catastrophe shifted expert opinion about the value of using OCVs for cholera prevention and control (I_2).

As interviewees I_2 and I_6 described:

> "The reason why [OCVs] also became quickly accepted and used was related to the big cholera outbreak in Haiti. It was very important in pushing the importance for the cholera vaccine. Because earlier, the WHO had been very hesitant about introducing OCVs" (I_2)

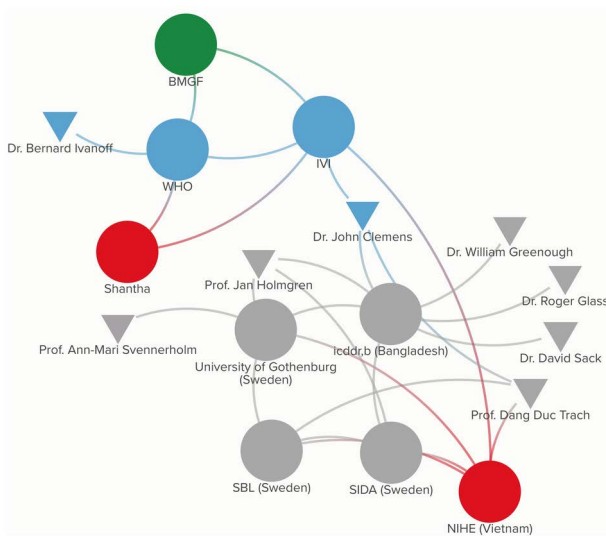

**Fig 6. Participatory network map for Shanchol.**

*"There was a big push by many individuals, including Paul Farmer and his team at Partners in Health. They held a famous phone call of the various cholera people around…When he heard cholera vaccines were available, and that people were dying of cholera in Haiti, his first question was, 'why are we not giving a vaccine like for any other outbreak?' When individuals in the more traditional public health sphere said, 'let's just push forward with hydration, because if you die of cholera, it's from dehydration' and 'let's push forward with improved sanitation, water supplies, etc.', he said, 'I'm not opposed to any of that, but we need to push forward also with as many tools as we have, and this is a tool'. So it was at that point that the global task force for cholera control was reinvigorated, and vaccines came back onto the radar screen" (I_6)*

As interviewee I_6 described, the renewed sense of urgency caused by the Haiti outbreak and desire to take advantage of the newly available lower-priced OCV, led to three pivotal actions: 1) the renewal of the global task force on cholera control [43], 2) a WHO position paper which recommended the use of OCVs in endemic areas and in areas at risk for outbreaks [31], and 3) the establishment of the first ever global OCV stockpile in 2013 [44], which was supported by Gavi with a funding commitment of $115 million to fund the stockpile from 2014 to 2018 [31]. The WHO's position paper in 2010 was especially influential in enabling the creation of the stockpile, as for the first time, they changed their position on the use of OCVs and recommended them as a preventive measure in the short-to-medium term to complement long-term improvements in WASH practices, and also to help stop the spread of outbreaks to new areas even after outbreaks had already begun [45]. Professor Ann-Mari Svennerholm was the chairperson of the enteric vaccine group of the WHO at this time and played an important role in the WHO expert meeting in Geneva in 2010 which led to this policy change (I_2). IVI also played an instrumental role in developing an OCV demand forecast to help build the global cholera investment case, which they published in 2012 [31]. Additionally, in 2012, Médecins sans Frontières (MSF) funded and conducted a study in Guinea to demonstrate the short-term effectiveness of Shanchol under field conditions and found that Shanchol was in fact effective when used in response to a cholera outbreak [46]. This provided further evidence for the value of using OCVs in outbreak response and supported efforts to establish a global OCV stockpile.

Meanwhile, Sanofi began engaging in developing the cholera vaccine in 2011 after acquiring Shantha, the producer of Shanchol. During this time, Sanofi self-financed further improvements to some parts of the process and analytical methods, and they developed and licensed Shanchol's Extended Controlled Temperature Conditions (ECTC). Shortly after, in partnership with the WHO and the International Coordinating Group (ICG) on Vaccine Provision, Sanofi created the first stockpile (2 million doses) of OCVs for emergency response (I_16). MSF led the first deployment of vaccines from the stockpile to South Sudan in 2014, and thus highlighted the feasibility of stockpile use, while also stressing the importance of coordination between stakeholders involved in cholera prevention and control, and the need to grow the global OCV stockpile [47].

Once the global OCV stockpile was formally established, Sanofi became a regular supplier to the stockpile. Since receiving WHO PQ in 2011, around 35 million doses of Shanchol have been manufactured and delivered (I_16). However, in 2020, Sanofi announced their decision to end production of Shanchol and withdraw from the market by the end of 2023.

A representative from Sanofi explained that from the beginning, Shanchol had been challenging to produce due to its complex and suboptimal manufacturing process, which made it difficult to remain competitive. Furthermore, despite an adapted pricing strategy, the manufacturing costs of Shanchol led to a final production cost that was approximately twice

as high as that of other OCVs used in endemic markets. While Sanofi engaged in efforts to improve the product profile of Shanchol, they were ultimately unsuccessful. As Sanofi further described, Shanchol represented a fraction of the total number of doses produced globally (approximately 10% of the total cholera vaccines produced per year), and the leading manufacturer, EuBiologics, announced an increase in production capacity, while new manufacturers also announced their intent to enter the cholera vaccine market with high volumes and low prices. Therefore, in 2022, IVI and Sanofi signed a transfer of knowledge agreement granting IVI non-exclusive, worldwide, sub-licensable, free of charge access to the intellectual property rights relating to the manufacture of Shanchol (which had been further improved upon and developed by Sanofi), to support IVI in their efforts to bring on additional manufacturers to the cholera vaccine market (I_16).

### D. Euvichol

The BMGF's original grant to IVI required them to engage in technology transfer for the OCV with two manufacturers, to increase the global OCV supply and promote healthy price competition. The first technology transfer was conducted with Shantha for the production of Shanchol, as previously mentioned. For the second technology transfer, IVI had contacted various members of the Developing Countries Vaccine Manufacturers (DCVMs) Network International for technology transfer and product development partnerships, but most showed little interest due to the lack of commercial incentives for cholera vaccines [31]. At the time, EuBiologics was a small Contract Research and Manufacturing Organization (CRMO) based in the Republic of Korea (ROK), that had no prior experience in vaccine production. However, the founders of EuBiologics had abundant experience in production, quality, and regulatory affairs with major vaccine companies based in the ROK. Interviewee I_11 explains the rationale behind EuBiologics submitting a proposal to IVI and how the collaboration came to be:

> "The OCV is an affordable vaccine mainly targeting low- and middle-income countries (LMICs). So, the big pharmaceutical companies in South Korea were not interested in developing affordable vaccines for LMICs. [EuBiologics] thought that they would have a chance to collaborate with IVI, and if they decided to engage in tech transfer with EuBiologics, then EuBiologics could have investments from external parties to set up manufacturing facilities for the development of the OCV…also, it's a small company, which means that the decision-making timeline is really quick, and they can bring the product to the market quickly. Also, since IVI and EuBiologics are both based in South Korea, they can collaborate together. So, EuBiologics got positive feedback from the BMGF, then IVI decided to transfer technology to EuBiologics"(I_11)

In October 2010, IVI facilitated the technology transfer process from Shantha/Sanofi to EuBiologics, which was a time and resource intensive process for both manufacturers. Interviewee I_1 further described the role IVI played during the technology transfer process stating:

> "EuBiologics was IVI's second tech transfer. That is a company that itself had never done any vaccine manufacturing before. So even though they did it in their facility, IVI provided a lot of assistance beyond the technical scale up, like how to get financing, how to interact with the global world that's interested in cholera... So, IVI is an adjustable facilitator, it can help with the technical aspect, or with navigating and integrating companies into the complex global ecosystem in order to be commercially successful" (I_1)

According to interviewee I_10, the development of Euvichol from inception to WHO PQ took nearly five years and cost a total of 19.7 million USD [31]. In addition to the technology transfer process which was financially supported by IVI through the BMGF grant, EuBiologics secured funding for the development of Euvichol from a diverse group of funders, who are summarized in Table 2.

At this time, investors had met with the BMGF and had been discussing the possibility of Gavi providing funding to support the stockpile. Once the funding commitment was confirmed, EuBiologics received their first investments from venture capitalists in 2013, which helped them to purchase capital equipment and supported the final clinical studies and regulatory processes for Euvichol [31].

Next, EuBiologics enlisted the help of universities in the ROK to produce clinical trial materials for the phase 1 clinical study they conducted in the ROK. An additional phase 1 clinical study was required by the Korean Ministry of Food and Drug Safety (MFDS) as it was the first study conducted with clinical trial material produced by EuBiologics. After generating successful phase 1 results, EuBiologics was able to begin phase III trials to assess the safety and immunogenicity of their vaccine as compared to Shanchol [31]. To produce the vaccine, EuBiologics outsourced the manufacturing to the Green Cross Corporation (who was also one of their investors) (I_11). As there were no cholera cases within the ROK, the MFDS would not approve the phase 3 clinical trials being conducted in Korea, but rather, they agreed to review and approve an "export only" license which had to be evaluated outside of the ROK. IVI was instrumental in facilitating the phase III clinical trial in the Philippines, as they organized a meeting between EuBiologics, the MFDS, and the WHO PQ team. In addition, the clinical trial experts at IVI lent EuBiologics their support and expertise throughout the clinical trials process. In 2014, the clinical study was finished and demonstrated that two doses of the EuBiologics vaccine had comparable immunogenicity to Shanchol [31].

In January 2015, the MFDS officially approved the vaccine for export-only licensure, and with the support of IVI, Euvichol received WHO PQ in late 2015 [31]. Shortly thereafter, EuBiologics changed the presentation of Euvichol from a glass vial to a plastic tube, which decreased its weight and volume, further facilitating shipping, storage, transportation, and reduced wastage; the new vaccine, Euvichol-Plus, obtained WHO PQ in August 2017 [31] (I_12). Fig 7 presents the participatory network map for Euvichol-Plus.

In regard to the intellectual property agreement associated with Euvichol-Plus, IVI managed the know-how for the OCV technology and licensed it to EuBiologics. Interviewee I_11 explained that there were several conditions attached to this licensing agreement:

Table 2. Overview of Euvichol investors [31].

| Investor name | Description | Percentage of funding |
|---|---|---|
| Korea-Seoul Life Science Fund | Cross-border life sciences VC fund founded in ROK. | 20% |
| Green Cross Corporation | Biopharmaceutical company headquartered in ROK. | 8% |
| Shinhan-K2 Investment Partners | Growth-stage VC fund headquartered in ROK. | 6% |
| Global Health Investment Fund (sponsored by the BMGF) | Social impact investment fund designed to provide financing to advance the development of drugs, vaccines, diagnostics, and other interventions against diseases that disproportionately burden LMICs. | 29% |
| Korea Investment Global Frontier Fund | Venture capital fund managed by Korea Investment Partners. | 18% |
| EuBiologics | Self-funding | 19% |

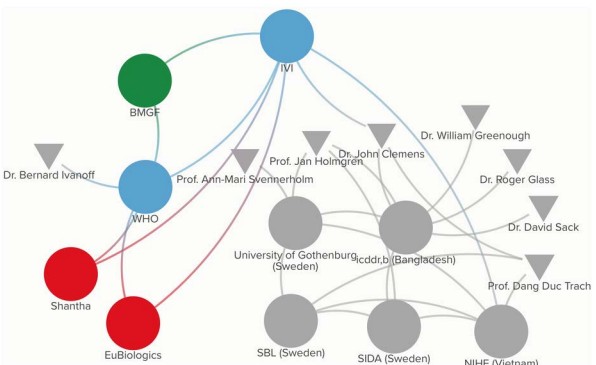

**Fig 7. Participatory network map for Euvichol-Plus.**

> *"EuBiologics needed to have manufacturing capacity, at least up to 6 million doses of production per year. Then they should be able to produce 15 batches in compliance with WHO guidelines for PQ, and the vaccine pricing itself should be between $1 and $2. Those were the conditions that EuBiologics had to satisfy when they executed their licensing agreement with IVI" (I_11)*

Interviewee I_1 expanded upon IVI's approach to managing intellectual property for the OCV, explaining:

> *"The way OCV developed in academia, there were a sufficient number of early publications about the process from Jan Holmgren, so it really wasn't patentable anymore because so much had been made public. But what IVI licenses is know-how and the [inactivated bacterial] strains…IVI shares with the licensee the know-how which trains them and all the documentation, and then the strains, which IVI controls – is a really huge deal. Could someone else just make a similar vaccine using different strains? Absolutely. But they'd have a much longer regulatory process to go through. Whereas, because these strains have been in millions of people and are recognized as safe, when IVI does a new tech transfer, the new companies don't have to go through that process. So, the cost for someone to copy and do something using different strains would just be too high. So, in that way IVI controls it, and thereby ensures the cost affordability. Why does IVI want to control it? Because they want there to be a global access agreement. If anyone does this, IVI's mission is to make sure it's done at low cost" (I_1)*

Euvichol-Plus is currently priced at $1.30 per dose for procurement by Gavi for the global OCV stockpile (I_12). Initially, EuBiologics suggested a price for Euvichol based on a cost-plus formula estimate, but soon revised it after the BMGF assessed that the price was too low for the organization to remain sustainable, and instead suggested that EuBiologics offer scaled discounts based on volume of units sold (I_9).

Interviewee I_4 explained the BMGF's desire for EuBiologics to raise their price:

> *"Since this was the only product they had, the BMGF needed to ensure their success by making sure that they had a little bit more profit. The BMGF also thought that they were a bit naive in terms of what they were offering. The BMGF gave EuBiologics a grant in 2020 to expand their facility, and then renegotiated the whole global access agreement. At this point,*

*the Foundation had refined its policy. So instead of leaving it to the partner, they are now intimately involved…It's a protracted negotiation. The global access and commercialization agreements involved projecting likely demand forecasts for the vaccine and volume-based pricing to keep the costs competitive and the company sustainable. This is based on the premise that the BMGF needs and wants them to be successful as a company and aligns with their mission as a charitable organization" (I_4).*

Interviewee I_11 explained that this volume-based pricing strategy enables EuBiologics to cover their fixed organizational costs, maintain their facility, and remain sustainable, while providing international procurement agencies with affordable prices which they can also adjust as needed to the demand which materializes (I_11). As interviewee I_12 also explained:

*"EuBiologics' pricing is really competitive. They have a [volume based] pricing system, so the highest pricing is $1.65, but it goes down as they supply more volumes. So, the weighted average pricing is about $1.25. So, it's really affordable pricing. Their margin is capped at 30%, and they have a collaborative access agreement in place with the BMGF and IVI, so they cannot increase the pricing" (I_12)*

EuBiologics' revenue from the sales of Euvichol-Plus currently accounts for more than 90% of their total revenue, and their demand comes almost entirely from international procurement agencies (I_12). EuBiologics' annual royalty payment to IVI is $10,000 USD. If EuBiologics happens to sell more than $10,000 USD worth of Euvichol-Plus on the private market in any year, then as part of their agreement, IVI would receive an additional royalty payment of 5% of the additional amount (I_9). Finally, EuBiologics has also made milestone payments to IVI, such as the payment they made for $50,000 upon completion of technology transfer training (I_12).

IVI facilitated a long-term purchase agreement for Euvichol from 2016 to 2018 to supply the global OCV stockpile [31]. The agreement has since been renewed, and EuBiologics is currently the largest supplier of the OCV in the world, producing 35 million doses annually (prior to the WHO PQ of Euvichol-S, which has since prompted an increase in their production capacity) [48] (I_12).

## E. Euvichol-S, DuoChol, Hillchol, CCV, and additional tech transfers

In addition to the existing OCVs, there are currently four additional cholera vaccines that have since become available or are currently under development, which have been created by stakeholders within the existing cholera vaccine ecosystem discussed in this paper – Euvichol-S, DuoChol, Hillchol, and a cholera conjugate vaccine, as well as additional technology transfers facilitated by IVI.

**Euvichol-S.** Motivated by the desire to decrease the complexity of Euvichol-Plus and thus increase production capacity, Euvichol-S is a simplified formulation of the Euvichol-Plus vaccine, developed by EuBiologics in collaboration with IVI. Together, they reformulated Euvichol-S by reducing the components from five bacterial strains to two, and the inactivation process from two processes to only one, which was estimated to reduce production costs by ~20% and increased production capacity by ~38% [49]. The reformulation process was funded by the BMGF [49]. Successful phase III clinical trials conducted by IVI demonstrated that Euvichol-S was non-inferior to Shanchol, and Euvichol-S was licensed for export by the Korea MFDS on December 19, 2023 [49]. On 12 April 2024, Euvichol-S received WHO PQ [16]. With the addition of Euvichol-S, EuBiologics now plans to produce 52 million doses of OCV

in 2024, including 15 million doses of Euvichol-S, while increasing its overall manufacturing capacity to 90 million doses annually [49].

**DuoChol.** In collaboration with Holmgren's team at Gothenburg, IVI began clinical development of DuoChol, which is a low-cost OCV with a similar composition to Dukoral but is a dry formulation in capsule form [28]. Funding for the development of DuoChol has been provided by the Wellcome Trust and the Swedish government [28]. IVI explained that the presentation in capsule form improves the vaccine's thermostability, thus the active ingredients remain stable at higher temperatures for a longer period, while also reducing its weight and volume [28]. However, the vaccine would still require a buffer because of the recombinant cholera toxin B-subunit and will not be suitable for young children.

**Hillchol.** Hillchol is a novel single-component OCV created by Holmgren's team at Gothenburg, and then tech transferred and further developed in collaboration with Hilleman Labs. The technology for Hillchol was licensed to Bharat Biotech, which also conducted a Phase III clinical trial. Due to its unique formulation, Hillchol can be produced at a lower cost and in larger quantities than any of the other existing OCVs today. Funding for Hillchol was provided directly by the SIDA and BMGF, and indirectly via a grant from the Wellcome Trust to Hilleman Labs. Hillchol obtained national licensure in India in August 2024 and is in the process of seeking WHO PQ (I_2).

**CCV.** As the existing OCVs are less efficacious in children between 0–5 years of age, development is currently underway for a cholera conjugate vaccine (CCV) that would have improved efficacy for children between the ages of 0–5, and also have a longer duration of protection [50]. IVI, EuBiologics, and a research team from Massachusetts General Hospital (MGH)/Harvard are collaborating on the CCV, and have secured funding from Wellcome Trust, the RIGHT Foundation, and Open Philanthropy. Initial product development was supported through the U.S. National Institutes of Health. Next, the RIGHT Foundation provided a grant to EuBiologics with co-funding from Wellcome Trust to fund the tech transfer process for the CCV from MGH to EuBiologics (I_14). Currently, a phase 1 clinical trial for the CCV is underway, which has been supported by a RIGHT Foundation grant to IVI, with co-funding from Open Philanthropy (I_14) [51,52].

**Additional tech transfers.** In addition to the new formulations and technologies of cholera vaccines under development, IVI has also engaged in technology transfer with four additional manufacturers in an effort to meet national supply goals and also increase the number of suppliers to the global OCV stockpile. The manufacturers include Incepta Vaccine Ltd in Bangladesh, Bharat Immunologicals and Biologicals Corporation Limited (BIBCOL) and Biological E Ltd. in India, and Biovac Institute in South Africa. The vaccine "Cholvax" manufactured by Incepta Vaccine Ltd has been licensed nationally in Bangladesh [53], while the BIBCOL, Biological E Ltd., and Biovac vaccines are currently under development. Fig 8 presents the participatory network map for Euvichol-S, DuoChol, Hillchol, CCV, and additional tech transfers.

**The implications of a global donor-funded market.** As witnessed by EuBiologics' experience with Euvichol, the creation of the global OCV stockpile acted as an important source of pull funding that guaranteed an OCV market for EuBiologics once their vaccine received WHO PQ, enabling them to attract private investors. However, having the global OCV market effectively controlled by Gavi in the form of the stockpile has posed several implications for manufacturers. For instance, there is limited funding available from Gavi to purchase OCVs for the stockpile [12], likely due to an overall scarce budget and competing priorities, which directly affects the market size. Furthermore, the International Coordinating Group (ICG) on Vaccine Provision and Gavi's Independent Review Committee (IRC)

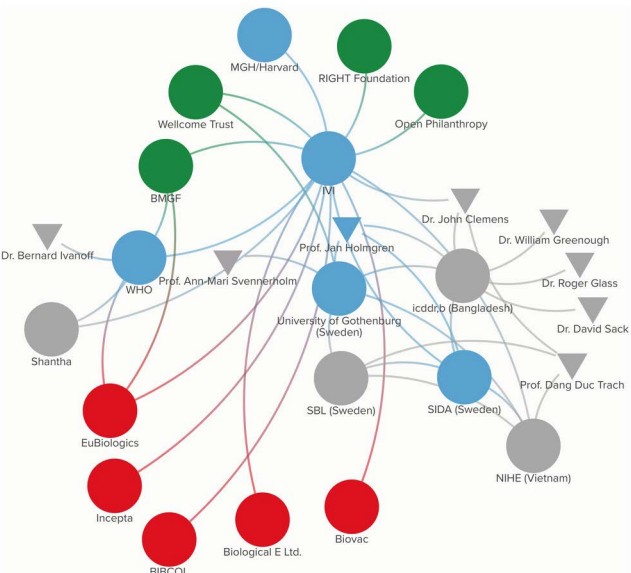

**Fig 8. Participatory network map for Euvichol-S, DuoChol, Hillchol, CCV, and additional tech transfers.**

determine the number of doses that are able to be distributed from the stockpile – the ICG reviews and approves requests for doses designated for outbreak response, and the IRC manages requests for preventive immunization campaigns. These unique market dynamics have potentially contributed to the global cholera vaccine shortages experienced today. For example, despite the fact that there are currently 40 million doses available per year via the stockpile, demands from countries have exceeded supply two- to threefold (I_2). Experts believe that the true global demand for OCVs is about 150 million doses per year, when accounting for vaccines to be used in both existing cholera outbreaks and in preventive vaccination campaigns. Thus far, most of the available vaccine doses have been exclusively approved for use in existing outbreaks which has stifled the use of OCVs for preventive campaigns in cholera hotspots, as envisioned in the GTFCC's Roadmap for Ending Cholera by 2030 (I_2).

As interviewee I_12 explained:

*"It's a donor-funded market. So, the GAVI roadmap and their policy changes are really important since they affect whether they decide to make investments in procurement for OCVs. I think it was in 2018, Gavi decided to fund operational costs of the vaccination campaigns of $0.65 per dose, and demand has increased since then. So, the Gavi funding and policies largely shape the market. It's not like other markets" (I_12)*

As a result, the potential market size limits the number of manufacturers who are willing to enter the market. As of 2024, EuBiologics is the sole supplier of the global OCV stockpile and is attempting to meet all existing demand.

As interviewee I_12 described:

*"There's less incentive for other manufacturers to enter the market because they know that the market is really small. For EuBiologics, the revenue they generated last year was about $40 million, and their market share represents 90% (with Sanofi accounting for the other*

*10%) which means that the public market as a whole is about $50 million. And then demand is really uncertain. It's mostly in outbreak settings or campaigns. So, if the vaccine is included in routine immunization, then it is very easy to calculate the demand, but it's in preventive campaigns and outbreaks, so the demand is really uncertain. Also, we say that we're in a supply shortage, but it's been one or two years that OCV is in a supply constrained market while EuBiologics has excess capacity in their manufacturing site" (I_12)*

Moreover, due to the characteristics of cholera as a disease, demand can be very volatile. While some countries can be declared cholera-free for years, a sudden outbreak can spread uncontrollably, and events connected with climate change are causing cholera outbreaks to increase globally. In addition to the unpredictability of outbreaks, interviewees explained that the stigma and political dimensions of reporting cholera outbreaks also pose difficulties. Governments often report cases as "acute watery diarrhea" to avoid the economic repercussions of potential trade and travel bans targeting directly affected countries when cholera outbreaks are reported (I_8). Together, these characteristics make it increasingly difficult for UNICEF to create accurate demand forecasts, and for suppliers like EuBiologics to produce an appropriate amount of supply. These challenges are similar to those many product development partnerships (PDPs) experience for neglected disease products – in the absence of a clear buyer for products, products are liable to fall into a gap and not reach their intended users.

**A collaborative network innovation model.** The evolution of OCVs over a more than 30-year trajectory demonstrates that a collaborative network innovation model in which a few key actors drive end to end product development, has the potential and ability to successfully develop new pharmaceutical products that are affordable and well-adapted for use in resource-poor settings. The OCV story emphasizes the power of interpersonal relationships – a group of stakeholders who knew and trusted one another, helped each other to secure knowledge, funding, and manufacturing resources. Their sustained, collective efforts for over three decades enabled them to continue expanding production and improving OCVs from one iteration to the next. Additionally, it enabled them to change public opinion and convince the WHO to recommend the use of cholera vaccines as another valuable tool for prevention and control. Moreover, they were able to promote a donor-funded market in the form of the global OCV stockpile to overcome some of the traditional challenges associated with the neglected diseases and pandemic preparedness markets and incentivize the development of OCVs.

Fig 9 depicts the collaborative network innovation model in broad terms, and lists each of the elements used to build the innovation model, differentiating between those elements which can be considered mainstream versus alternative.

Each of the elements used for the innovation model pose implications which affected the development of the OCVs discussed, and for collaborative network innovation models to be used for other disease areas moving forward. First, in terms of funding, public and philanthropic funding is most readily available for diseases with limited profit potential such as cholera, and the flexibility this funding provides can be particularly valuable. However, the lack of private funding leads to a precarious market situation and makes it difficult for vaccine development to progress smoothly from one stage of the R&D process to the next, without getting stalled while developers wait to secure additional philanthropic and public funding. Moreover, because philanthropic funding is the majority of the funding available, funders have significant influence over the development process.

In terms of knowledge, within this innovation model stakeholders benefit more significantly from existing prior and internal knowledge that is passed from one individual or entity to the next. In addition, the increased use of technology transfer from one developer to the

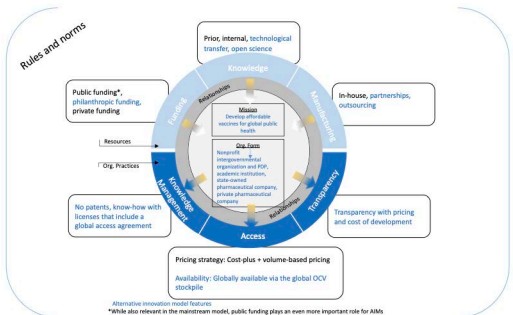

**Fig 9. Collaborative network innovation model.**

next helps to significantly de-risk the development process, while also greatly shortening the entire R&D timeline. Together, these actions help to reduce the traditional costs and resources associated with mainstream pharmaceutical development. Beyond technical expertise, internal knowledge related to other aspects of the development process such as navigating regulatory processes and securing funding, are also valuable resources that make the pharmaceutical development process more efficient and successful, particularly for more nascent organizations.

In terms of manufacturing, while in-house manufacturing exemplifies more of the mainstream innovation model, the collaborative network innovation model relies more heavily on partnerships and outsourcing for manufacturing processes. Therefore, forming strong, reliable partnerships with external manufacturers can also help pharmaceutical developers to find opportunities to reduce certain manufacturing costs or make processes more efficient.

The knowledge management practices illustrated by vaccine developers in the collaborative network innovation model for OCVs were also particularly unique and helped to achieve global access. Holmgren never patented his original technology for Dukoral, and instead published his data freely and openly via scientific publications. IVI also never patented any of their reformulations or processes, but rather took ownership of the know-how and the strains used in their vaccines and licensed this know-how and these particular strains to the manufacturers they collaborated with. Therefore, IVI effectively controlled this knowledge even in the absence of a patent, and thereby ensured affordability through their associated global access agreements with companies. Furthermore, IVI's collaboration with each of the manufacturers who supplied the global OCV stockpile enabled them to ensure the pricing strategy used for these OCVs successfully balanced affordability for international procurement agencies with sustainability of the organizations and were globally accessible through their global access agreements. Finally, the majority of OCV developers were also transparent in terms of publishing their pricing and costs of development.

## IV. Discussion and conclusions

The OCV case study demonstrates that interpersonal relationships of trust and collaboration are arguably the most important resource for those involved in vaccine development that do not have the same access to resources as large pharmaceutical companies do. As evidenced by each OCV's participatory network map, Jan Holmgren's initial vaccine technology has been tangentially linked to each of the next iterations of OCVs, from mORC-Vax and Shanchol to Euvichol-Plus, and now Euvichol-S. In addition, the relationships formed between Holmgren at University of Gothenburg, Trach at VaBiotech, and Clemens and Glass at icddr,b, paved the way for OCVs to be put on the global agenda, and gain support from WHO. Also, Clemens'

appointment as the first Director-General of IVI directly led to IVI's involvement in facilitating the development of OCVs for purchase by international procurement agencies.

However, as evidenced by the current mismatch between OCV supply and demand, the collaborative network innovation model can pose structural weaknesses which arise from such a small group of people controlling so many aspects of the market, from vaccine development and manufacturing to purchasing. Therefore, there needs to be better coordination between the stakeholders responsible for producing, purchasing, and distributing OCV doses. Related to this, a firm demand for cholera vaccines needs to be established in advance, to incentivize manufacturers to manufacture volumes of OCVs at-risk in the absence of advance purchase guarantees. Better demand forecasting and supply planning will be essential for OCVs as cholera outbreaks will continue and thus, shortages will otherwise persist.

Most significantly, the OCV story provides evidence for the feasibility of a collaborative network innovation model to be implemented in other disease areas, such as pandemic preparedness for instance. Such models have been established for Covid-19, such as the COVID Moonshot initiative [54], which has been considered a new and untested open science academic network innovation model for the development of COVID-19 therapeutics, or the Rapidly Emerging Antiviral Drug Development Initiative (READDI) [55] for a broader set of emerging infectious diseases. Yet many parallels can be drawn between these initiatives and the collaborative network innovation model used to develop OCVs. Therefore, when considering the use of this type of alternative innovation model, several potentially applicable conclusions can be drawn from the successful development of OCVs.

First, the construction of broader collaborative networks appears to be a crucial element of developing drugs and vaccines for neglected diseases or outbreaks. Formalizing these networks and creating dedicated time and space for convening, such as the Global Task Force on Cholera Control does, is essential to facilitate the R&D process, specifically for disease areas that lack market incentives. These interpersonal relationships which form between individuals significantly help them to access the resources necessary for vaccine development.

In terms of specific resources, philanthropic and public funders are most suited to fund drug development in areas which lack market incentives. As evidenced by the role of the BMGF in OCV development, funders also have the ability to greatly incentivize developers to enter the market and can directly help get drugs developed for diseases that have no existing treatment or improve upon existing drugs to be better suited for resource-constrained settings. Therefore, facilitating collaboration between funders with these common interests is essential. In addition, funders facilitated the sharing of existing prior and internal knowledge among actors, including ensuring technology transfer. Together, these actions can help to increase the number of manufacturers or suppliers for any given product, and thus, their ability to meet global demand, which is particularly relevant within the pandemic preparedness niche. In addition to technical knowledge, existing knowledge related to how to navigate other aspects of the R&D process such as regulatory and financing can also help stakeholders progress more efficiently and effectively, especially for nascent organizations (as demonstrated by EuBiologics' experience with the help of IVI). Finally, utilizing alternative knowledge management practices, such as implementing licensing agreements with global access terms instead of patents, helps to ensure that products remain globally affordable and accessible while continuing to increase the number of manufacturers who can produce the product.

International agreements, such as the WHO pandemic treaty currently under negotiation, could act as valuable opportunities to formalize these collaborative network innovation models on a global scale in disease areas such as pandemic preparedness. This would help facilitate the development of vaccines for future pandemics. Within this type of international agreement, having a global pandemic preparedness task force composed of actors from the

Global North and South who have the requisite knowledge, expertise, and desire to develop drugs for pandemic preparedness would help facilitate the R&D process. Moreover, creating a new pandemic preparedness funding mechanism could similarly create a market for vaccines for future pandemics, and aid in stockpiling efforts, such as the global OCV stockpile has. In addition, public funding for R&D and facilitating access to prior knowledge can replace the need for patents as an incentive for innovation; attaching global access conditions to such funding and ensuring technology transfer could increase the number of suppliers and also promote vaccine equity, particularly for low- and middle-income countries. The successful development of OCVs sustained over more than three decades illustrates the power of inter-personal relationships and the collective efforts of individuals dedicated to making a difference in their disease area of expertise. This collaborative network innovation model can and should be replicated to successfully develop future products for global public health.

## Acknowledgments

We would like to express our gratitude to each of the interviewees who generously agreed to participate in the research and provided their time and unique perspectives into the development of each cholera vaccine, which was instrumental to telling this complex story. We would also like to thank them for the helpful feedback they provided on the draft of this manuscript. We would also like to thank Dr. Adam Strobejko for the feedback and edits he provided for this manuscript, and Erika Shinabargar for her assistance with transcribing interviews.

## Author contributions

**Conceptualization:** Kaitlin Large, Adrian Alonso Ruiz, Iulia Slovenski, Marcela Vieira, Suerie Moon.

**Data curation:** Kaitlin Large.

**Formal analysis:** Kaitlin Large.

**Funding acquisition:** Suerie Moon.

**Investigation:** Kaitlin Large.

**Methodology:** Kaitlin Large.

**Project administration:** Adrian Alonso Ruiz, Suerie Moon.

**Resources:** Suerie Moon.

**Supervision:** Suerie Moon.

**Validation:** Kaitlin Large, Adrian Alonso Ruiz, Iulia Slovenski, Marcela Vieira, Suerie Moon.

**Visualization:** Kaitlin Large.

**Writing – original draft:** Kaitlin Large.

**Writing – review & editing:** Kaitlin Large, Adrian Alonso Ruiz, Iulia Slovenski, Marcela Vieira, Suerie Moon.

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
