## [Decision Letter · Decision Letter 0]

9 May 2024

PGPH-D-24-00562

The 30-year evolution of oral cholera vaccines: A case study of a collaborative network alternative innovation model

Dear Dr. Large,

Thank you for submitting your manuscript to PLOS Global Public Health. After careful consideration, we feel that it has merit but does not fully meet PLOS Global Public Health’s publication criteria as it currently stands. Therefore, we invite you to submit a revised version of the manuscript that addresses the points raised during the review process.

We look forward to receiving your revised manuscript.

Kind regards,

Syed Shahid Abbas, MBBS, MPH, Ph.D.

Academic Editor

Journal Requirements:

Additional Editor Comments (if provided):

Thank you for sharing your manuscript for consideration at PLOS Global Public Health. Please consider both the reviewers' comments about clarifying the discussion of your findings. Is it also possible to provide the 'work-in-progess' references you have cited for reviewers to consider, or, alternatively, change them to publicly available documents?

Reviewers' comments:

Reviewer's Responses to Questions

**Comments to the Author**

1. Does this manuscript meet PLOS Global Public Health’s publication criteria ? Is the manuscript technically sound, and do the data support the conclusions? The manuscript must describe methodologically and ethically rigorous research with conclusions that are appropriately drawn based on the data presented.

Reviewer #1: Yes

Reviewer #2: Yes

2. Has the statistical analysis been performed appropriately and rigorously?

Reviewer #1: N/A

Reviewer #2: N/A

3. Have the authors made all data underlying the findings in their manuscript fully available (please refer to the Data Availability Statement at the start of the manuscript PDF file)?

Reviewer #1: Yes

Reviewer #2: No

4. Is the manuscript presented in an intelligible fashion and written in standard English?

Reviewer #1: Yes

Reviewer #2: Yes

5. Review Comments to the Author

Reviewer #1: This is a well written paper and the research is strongly conducted. I really enjoyed reading it. It is an interesting story. It is great to document how vaccine development has been achieved by only a few individuals and how important collaborations are. I only have minor comments. I would say that my recommendation is between major and minor revision. It is not a lot that needs changing in my opinion.

1. Page 5, lines 108-109: It is not acceptable to reference papers in preparation as these cannot be looked up by the reader. There are indeed already published papers on drug development within neglected disease. Please reference some of these instead as that would be useful for the reader to know about.

2. Please check through that all abbreviations are spelled out the first time you mention them. Lines 153 and 154: I don't think we define IVI and ROK

3. Lines 151-154. Please state how many interviews were undertaken with each of the mentioned methods. How many on video, how many in person?

4. Line 174: The analytical framework is mentioned as 'developed for the broader research project'. Is this the same as CAPIS mentioned earlier? It is necessary to describe the framework in this paper as this is what you use for data analysis. Please add this.

5. Table 1: In the column on 'Developers', please add the country the developers are based in. This is so it is easier for the reader to align the figure and the table - and easier to remember who is who

6. Line 258: SIDA is the Swedish Government Development Agency and not something you can be a 'member' of. Was he a staff at SIDA? Or does SIDA have an advisory Board?

7. Line 336: Gavi was established in year 2000, so it is confusing that it is mentioned already here in a section about 1997. The stockpile of course came much later.

8. Line 561: Unclear what 'drug product' is being referred to since the topic is vaccines

9. It is stated in line 511 that IVI received grant from BMGF to organize two technology transfers. You only describe the one to to EuBiologics. Please clarify if they are working on another one as part of this agreement.

10. Lines 567-569: Please explain why a phase III trial is necessary for EuBiologics. Is a phase III trial necessary when it is tech transfer of a vaccine where efficacy has already been shown? You mention in line 569 that the trial showed similar immogenicity to Shanchol. I don't think that a phase III trial is needed to show immunogenicity. is that not only a phase II?

11. Line 640: 'Currently' is repeated twice

12. You cannot include new findings in the Discussion and Conclusion section. it is not recommended to include a figure in this section. The figure and the accompanying text should be moved to the results section. This will also make the discussion an appropriate lenght. It is too long as it is. I recommend that the discussion starts from line 814.

13. Lines 837-842: Please add references for Moonshot initiative and READDI. I cannot be assumed that the reader know about this.

Reviewer #2: Thank you for the opportunity to review this interesting article "The 30-year evolution of oral cholera vaccines: A case study of a collaborative network alternative innovation model.” The manuscript tells the story of the long road to our current oral cholera vaccines and provides a great perspective on how other vaccines for diseases of the poor could proceed. This story will be a nice contribution to the literature and I only have a couple ‘major’ comments along with several minor suggestions/questions.

Major Comments:

The participatory network maps are really wonderful but I think it would be helpful for the authors to provide more guidance on how to interpret them (e.g., do edge lengths matter? coloring of edges? size of nodes? placement of nodes?). Can we assume that these maps capture all the direct transfers of funding between donors and recipient? For example in Figure 7 BMGF only gave money directly to WHO and IVI? Wellcome to IVI and U. of Gothenburg? I think the authors should double check each of these (some small comments below) as think there may be some omissions/mistakes.

The actual use of OCV in emergencies and outbreaks played an important role in the story yet I feel like this was largely ignored. The first use of the Dukoral in a humanitarian setting was with Dukoral in 1999 (https://www.ncbi.nlm.nih.gov/pmc/articles/PMC2557739/pdf/10593032.pdf ) which showed feasibility. It is important to note that those involved in this original deployment went on to lead the WHO Cholera Team at WHO (Dominique Legros) who helped to create a policy environment ripe for development and which opened the door to the ICG processes. Around the same time as Shanchol was being PQ’d MSF (Doctors without Borders) was the first to use this vaccine reactively (https://www.nejm.org/doi/full/10.1056/NEJMoa1312680 ) using thier own funds. MSF has a seat on the ICG and this use paved the way to greater acceptance of the potential role of reactive vaccine use. The first deploymenbt of the stockpile was in South Sudan (https://journals.plos.org/plosmedicine/article?id=10.1371/journal.pmed.1001901 ) again led by MSF. I suspect that this part of the story is missing partly because of the snowball sampling used (which can lead you to stay within a tight network) and strongly urge you to consider talking to others involved in this side of the story.

Minor Points

- The introduction states that incidence of cholera outbreaks has been rapidly increasing and this is largely due to population displacement caused by conflicts and climate change. While this is a convenient narrative (and something that WHO has more or less said), this is a scientific manuscript and I urge you to find evidence to support this statement or remove/tone it down. Both the ‘increase in outbreaks’ (not just changes in surveillance and willingness to report) and the link with climate change are debatable.

- L72 - I recommend you refer to the new surveillance guidelines for definition of an outbreak (https://www.gtfcc.org/wp-content/uploads/2023/02/gtfcc-public-health-surveillance-for-cholera-interim-guidance.pdf#page=11.07 )

- L75-78- The situation is a little more complex than suggested in the text. The ICG only controls the emergency stockpile of vaccines for outbreaks and other emergency situations. The rest of the OCV supply is purchased and controlled by Gavi (for use in multi-year vaccination plans that are typically considered preventive). This seems to be mentioned late in the manuscript (eg., L708) but that is a bit late.

- L87 - To be clear there has never really been a time when we didn’t have a shortage of OCV in the world since it has been PQd.

- L93-95 - Probably want to update now that Euvichol-S is PQd and being produced.

- L108-109 - I don’t think you can typically cite manuscripts in prep since no one can actually see/read these.

- L129-130 - Can you please add this as supplemental materials rather than asking people to request from author?

- L234 / 236 - Check references. Not sure what I_6 refers to.

- Figure 2: In lines 276:279 USAID is mentioned as a major donor yet they are not in Figure 2. Is this intentional?

- L291 - “autochthonous cases” meaning what exactly?

- L317 - mention the strain name.

- L355 - As far as I recall, the Mozambique study only measured outcomes in the first 4-5 months after vaccination. Also double check Zanzibar duration. Not sure it was longer than 1.5 years.

- Figure 5: what does the red outline of SIDA represent here? Also fro Figure 5, can you confirm that no funds went from Gates to Shanta directly?

- Table 3: It seems important to point out that the Global Health Investment Fund is something started by Gates. This now makes the foundation donors to this larger project and investors.

- Figure 6: I don’t see any direct links between IVI and Eubiologics here. Is this intentional? Also not clear to me what the colored edges between IVI —> U Gothenburg —> Eubiologics means (same for IVI —> Jan H. —> Shantha)?

- Line 601: Who is this the cost for? The stockpile or anyone?

- Since submitting this manuscript, Euvichol-S has been PQ’d and is going to be shipping soon. I suggest you rework the way Euvichol-S is discussed in light of this news.

- Table 1:

Might be useful to add a column indicating availability and/or stage of development. This table includes vaccines no longer produced and those not yet in clinical studies.

In the product characteristics there are data on efficacy for each. Are these from package inserts or generally from the 2021 review paper in Vaccine? I would suggest citing the original sources for the data for each instead. Do note that this is a hard comparison since OCV protection wanes with time.

Thank you!

Andrew Azman

6. PLOS authors have the option to publish the peer review history of their article (what does this mean? ). If published, this will include your full peer review and any attached files.

**Do you want your identity to be public for this peer review?** For information about this choice, including consent withdrawal, please see our Privacy Policy .

Reviewer #1: **Yes: ** Ulla Kou Griffiths

Reviewer #2: No

While revising your submission, please upload your figure files to the Preflight Analysis and Conversion Engine (PACE) digital diagnostic tool, https://pacev2.apexcovantage.com/ . PACE helps ensure that figures meet PLOS requirements. To use PACE, you must first register as a user. Registration is free. Then, login and navigate to the UPLOAD tab, where you will find detailed instructions on how to use the tool. If you encounter any issues or have any questions when using PACE, please email PLOS at figures@plos.org. Please note that Supporting Information files do not need this step.

---

## [Decision Letter · Decision Letter 1]

18 Dec 2024

PGPH-D-24-00562R1

The 30-year evolution of oral cholera vaccines: A case study of a collaborative network alternative innovation model

Dear Dr. Large,

Thank you for submitting your manuscript to PLOS Global Public Health. After careful consideration, we feel that it has merit but does not fully meet PLOS Global Public Health’s publication criteria as it currently stands. Therefore, we invite you to submit a revised version of the manuscript that addresses the points raised during the review process.

I agree with both the reviewers' comments about the authors' efforts to incorporate the reviewer comments. I would be happy to recommend this manuscript for publication after you address the clarification from Reviewer 2 (about HilChol vaccine being licensed in India)

We look forward to receiving your revised manuscript.

Kind regards,

Syed Shahid Abbas, MBBS, MPH, Ph.D.

Academic Editor

Journal Requirements:

Additional Editor Comments (if provided):

I agree with both the reviewers' comments about the authors' efforts to incorporate the reviewer comments. I would be happy to recommend this manuscript for publication after you address the clarification from Reviewer 2 (about HilChol vaccine being licensed in India)

Reviewers' comments:

Reviewer's Responses to Questions

**Comments to the Author**

1. If the authors have adequately addressed your comments raised in a previous round of review and you feel that this manuscript is now acceptable for publication, you may indicate that here to bypass the “Comments to the Author” section, enter your conflict of interest statement in the “Confidential to Editor” section, and submit your "Accept" recommendation.

Reviewer #1: All comments have been addressed

Reviewer #2: All comments have been addressed

2. Does this manuscript meet PLOS Global Public Health’s publication criteria ? Is the manuscript technically sound, and do the data support the conclusions? The manuscript must describe methodologically and ethically rigorous research with conclusions that are appropriately drawn based on the data presented.

Reviewer #1: Yes

Reviewer #2: Yes

3. Has the statistical analysis been performed appropriately and rigorously?

Reviewer #1: N/A

Reviewer #2: N/A

4. Have the authors made all data underlying the findings in their manuscript fully available (please refer to the Data Availability Statement at the start of the manuscript PDF file)?

Reviewer #1: Yes

Reviewer #2: Yes

5. Is the manuscript presented in an intelligible fashion and written in standard English?

Reviewer #1: Yes

Reviewer #2: Yes

6. Review Comments to the Author

Reviewer #1: All comments have been addressed well. I congratulate the authors for an excellent paper.

Reviewer #2: The authors did a nice job responding to the reviewer comments and I believe the manuscript is in good form to be published. Only one small thing to update before publication is that the HilChol vaccine is now licensed in India whereas the table in the manuscript says it has gone through a Phase II trial.

- Andrew Azman

7. PLOS authors have the option to publish the peer review history of their article (what does this mean? ). If published, this will include your full peer review and any attached files.

**Do you want your identity to be public for this peer review?** For information about this choice, including consent withdrawal, please see our Privacy Policy .

Reviewer #1: **Yes: ** Ulla Kou Griffiths

Reviewer #2: No

While revising your submission, please upload your figure files to the Preflight Analysis and Conversion Engine (PACE) digital diagnostic tool, https://pacev2.apexcovantage.com/ . PACE helps ensure that figures meet PLOS requirements. To use PACE, you must first register as a user. Registration is free. Then, login and navigate to the UPLOAD tab, where you will find detailed instructions on how to use the tool. If you encounter any issues or have any questions when using PACE, please email PLOS at figures@plos.org. Please note that Supporting Information files do not need this step.

---

## [Editor Report · Decision Letter 2]

31 Dec 2024

The 30-year evolution of oral cholera vaccines: A case study of a collaborative network alternative innovation model

PGPH-D-24-00562R2

Dear Ms. Large,

We are pleased to inform you that your manuscript 'The 30-year evolution of oral cholera vaccines: A case study of a collaborative network alternative innovation model' has been provisionally accepted for publication in PLOS Global Public Health.

Best regards,

Syed Shahid Abbas, MBBS, MPH, Ph.D.

Academic Editor